# Reactivation strength during cued recall is modulated by graph distance within cognitive maps

Simon Kern[1,2,3]*, Juliane Nagel[1,2,3], Martin F Gerchen[1,4,5], Çağatay Gürsoy[1,2,3], Andreas Meyer-Lindenberg[2,5], Peter Kirsch[1,4,5], Raymond J Dolan[6,7], Steffen Gais[8], Gordon B Feld[1,2,3,4]*

[1]Clinical Psychology, Central Institute of Mental Health, Medical Faculty Mannheim, University of Heidelberg, Mannheim, Germany; [2]Psychiatry and Psychotherapy, Central Institute of Mental Health, Medical Faculty Mannheim, University of Heidelberg, Mannheim, Germany; [3]Addiction Behavior and Addiction Medicine, Central Institute of Mental Health, Medical Faculty Mannheim, University of Heidelberg, Mannheim, Germany; [4]Department of Psychology, Ruprecht Karl University of Heidelberg, Heidelberg, Germany; [5]Bernstein Center for Computational Neuroscience Heidelberg/Mannheim, Mannheim, Germany; [6]Max Planck UCL Centre for Computational Psychiatry and Ageing Research, London, United Kingdom; [7]Wellcome Centre for Human Neuroimaging, University College London, London, United Kingdom; [8]Institute of Medical Psychology and Behavioral Neurobiology, Eberhard-Karls-University Tübingen, Tübingen, Germany

*For correspondence:
simon.kern@zi-mannheim.de
(SK);
Gordon.Feld@zi-mannheim.de
(GBF)

**Competing interest:** The authors declare that no competing interests exist.

**Abstract** Declarative memory retrieval is thought to involve reinstatement of neuronal activity patterns elicited and encoded during a prior learning episode. Furthermore, it is suggested that two mechanisms operate during reinstatement, dependent on task demands: individual memory items can be reactivated simultaneously as a clustered occurrence or, alternatively, replayed sequentially as temporally separate instances. In the current study, participants learned associations between images that were embedded in a directed graph network and retained this information over a brief 8 min consolidation period. During a subsequent cued recall session, participants retrieved the learned information while undergoing magnetoencephalographic recording. Using a trained stimulus decoder, we found evidence for clustered reactivation of learned material. Reactivation strength of individual items during clustered reactivation decreased as a function of increasing graph distance, an ordering present solely for successful retrieval but not for retrieval failure. In line with previous research, we found evidence that sequential replay was dependent on retrieval performance and was most evident in low performers. The results provide evidence for distinct performance-dependent retrieval mechanisms, with graded clustered reactivation emerging as a plausible mechanism to search within abstract cognitive maps.

## eLife assessment

This magnetoencephalography study reports **important** new findings regarding the nature of memory reactivation during cued recall. It replicates previous work showing that such reactivation can be sequential or clustered, with sequential reactivation being more prevalent in low performers. It adds **convincing** evidence, even though based on limited amounts of data, that high memory performers tend to show simultaneous (i.e., clustered) reactivation, varying in strength with item

distance in the learned graph structure. The study will be of interest to scientists studying memory replay.

## Introduction

Memory classically relies on three distinct stages: encoding (learning), consolidation (strengthening and transforming), and retrieval (reinstating) of information. New episodic memories are learned by encoding a representation, thought to be realized in a specific spatio-temporal neuronal firing pattern in hippocampal and neocortical networks (*Frank et al., 2000*; *Preston and Eichenbaum, 2013*). These firing patterns are reactivated during subsequent rest or sleep, sometimes in fast sequential sequences, a process linked to memory consolidation (*Born and Wilhelm, 2012*; *Feld and Born, 2017*). Similarly, during retrieval, the same firing patterns seen during encoding are replayed in a manner that predicts retrieval success (*Carr et al., 2011*; *Foster, 2017*). Even though replay has been studied most intensely with respect to the hippocampus, the replay of memory traces in temporal succession is suggested as a general mechanism for planning, consolidation, and retrieval (*Buhry et al., 2011*). While a rich body of evidence exists in rodents (*Ambrose et al., 2016*; *Chen and Wilson, 2023*; *Foster and Knierim, 2012*; *Ólafsdóttir et al., 2018*), the contributions of replay to memory storage and retrieval in humans are only beginning to be examined (*Brunec and Momennejad, 2022*; *Eichenlaub et al., 2020*; *Fuentemilla et al., 2010*; *Wimmer et al., 2020*).

One obstacle has been the difficulty in measuring sequential replay or general network reactivation in humans (here we follow the definition of *Genzel et al., 2020*, where *reactivation* is used as an umbrella term for any form of reoccurrence of a previously encoded neural pattern related to information-encoding, and *replay* refers to reactivation events with a temporally sequential nature). The most straightforward method is to use intracranial electroencephalography, though this is generally only feasible within individuals undergoing evaluation for the management of epilepsy (*Axmacher et al., 2008*; *Engel et al., 2005*; *Staresina et al., 2015*; *Zhang et al., 2015*). Another approach is to use functional MRI (*Schuck and Niv, 2019*; *Wittkuhn and Schuck, 2021*), though the latter is burdened by the challenge posed by the sluggishness of the hemodynamic response. Researchers have recently started to leverage the spatio-temporal precision of magnetoencephalography (MEG), in combination with machine learning-based brain decoding techniques, to reveal sequential human replay in humans across a range of settings that includes memory, planning, and inference (*Eldar et al., 2018*; *Kurth-Nelson et al., 2016*; *Liu et al., 2019*; *Liu et al., 2021b*; *McFadyen et al., 2023*; *Nour et al., 2021*; *Wimmer et al., 2020*; *Wimmer et al., 2023*; *Wise et al., 2021*). Many of the latter studies deploy a novel statistical analysis technique, temporally delayed linear modeling (TDLM) (*Liu et al., 2021a*). TDLM, and its variants, enables identification of sequential replay for previously experienced material during resting state (*Liu et al., 2019*; *Liu et al., 2021b*), planning of upcoming behavioral output (*Eldar et al., 2020*; *Kurth-Nelson et al., 2016*; *McFadyen et al., 2023*; *Wise et al., 2021*), and memory retrieval (*Wimmer et al., 2020*).

*Wimmer et al., 2020* reported sequential reactivation of episodic content following a single initial exposure during cued recall 1 d post encoding. Specifically, they showed participants eight short, narrated stories, each consisting of four different visual story anchor elements taken from six different categories (faces, buildings, body parts, objects, animals, and cars) and a unique ending element. In a next day recall session, participants were shown two story elements and asked whether both elements were part of the same story and whether the second element appeared before or after the first. At retrieval, they showed stories were replayed in reverse order to the prompt (i.e., when prompting element 3 and element 5, successful retrieval would traverse element 5 through 4 and arrive at element 3). However, this effect was only found in those with regular performance, while in high performers there was no evidence of temporal succession. Instead, the latter group simultaneously reactivated all related story elements in a clustered manner.

In memory research, declarative tasks often avail of item lists or paired associates (*Barnett and Blackwell, 2023*; *Cho et al., 2020*; *Feld et al., 2013*; *Kolibius et al., 2020*; *Roux et al., 2022*; *Schönauer et al., 2014*; *Stadler et al., 1999*; *Stadler et al., 1999*). When studying sequential replay, the task structure must have a linear element (*Liu et al., 2019*; *Liu et al., 2021b*; *Wimmer et al., 2020*; *Wise et al., 2021*) and such linearity is a defining feature of episodic memory (*Tulving, 1993*). By contrast, semantic memory is rarely organized linearly and instead involves complex and

interconnected knowledge networks or cognitive maps (*Behrens et al., 2018*), motivating researchers to ask how memory works when organized into a complex graph structure (*Eldar et al., 2020*; *Feld et al., 2021*; *Garvert et al., 2017*; *Schapiro et al., 2013*; for an overview, see *Momennejad, 2020*). However, little is currently known regarding the contribution of replay to consolidation and retrieval processes for information that is embedded in graph structures. In particular, the question remains how the brain keeps track of graph distances for successful recall and whether the previously found difference between high and low performers also holds true within a more complex graph learning context.

Here, we examined the relationship between retrieval from a learned graph structure and reactivation and replay in a task where participants learned a directed, cyclic graph, represented by 10 connected images. Eight nodes had exactly one direct predecessor and successor node, two *hub nodes*, each had two direct predecessors and successors (see Figure 5B). The task was arranged such that participants could not rely on simple pair mappings but needed to learn the context of each edge. Additionally, the graph structure was never shown to participants as a 'birds-eye view', encouraging implicit learning of the underlying structure. Following a retention period, consisting of 8 min eyes-closed resting state, participants then completed a cued recall task, which is the focus of the current study.

## Results

### Behavioral

All but one participant learned the sequence of 10 images embedded into the directed graph with partial overlap (*Figure 1—figure supplement 1*). On average, participants needed five blocks of learning (range, 2–6, see *Figure 1—figure supplement 2*) and attained a memory performance of 76% during their last block of learning (range, 50–100%). After 8 min of rest, retrieval performance improved marginally to a mean of 82% ($t = -2.053$, p=0.053, effect size $r = 0.22$; *Figure 1B*). Note that since the last learning block included feedback, this marginal increase cannot necessarily be attributed to consolidation processes. Additionally, we have included an analysis showing how wrong answers participants provided were random in the first block and biased toward closer graph nodes in later blocks. This is consistent with participants actually learning the underlying graph structure as opposed to independent triplets (see *Figure 1—figure supplement 3* for details).

### Decoder training

We first confirmed we could decode brain activity elicited by the 10 items using a cross-validation approach. Indeed, decoders were able to separate the items presented during the localizer task (see *Figure 1A*) well, with an average peak decoding accuracy of ~42% across all participants (range, 32–57%, chance level: 10%, excluding participants with peak accuracy <30%, for all participants; see *Figure 1—figure supplement 4*). We calculated the time point of the mean peak accuracy for each participant separately and subsequently used the average best time point, across all included participants, at 206 ms (rounded to 210 ms) for training of our final decoders. This value is very close in range to the time points found in previous studies (*Kurth-Nelson et al., 2016*; *Liu et al., 2019*; *Liu et al., 2021b*; *Wimmer et al., 2020*). The decoders also transferred well to stimulus presentation during the retrieval trials and could effectively decode the current prompted image cue with above-chance significance (cluster permutation test, see *Figure 1D*).

### Sequential forward replay in subjects with lower memory performance

Next, we assessed whether there was evidence for sequential replay of the learned sequences during cued recall. Using TDLM, we asked whether decoded reactivation probabilities followed a sequential temporal pattern, in line with transitions on the directed graph. Here, we focused on all allowable graph transitions and analyzed the entire time window, of 1500 ms, after onset of the retrieval cue ('current image'). We found positive sequenceness across all time lags for forward sequenceness, with a significant increase at around 40–50 ms state to state lag for forward sequenceness (*Figure 2A*). As discussed in *Liu et al., 2021b*, correction for multiple comparisons for this sequenceness measure across time is nontrivial and the maximum of all permutations represents a highly conservative statistic. Due to this complexity, we also report the 95% percentile of sequenceness

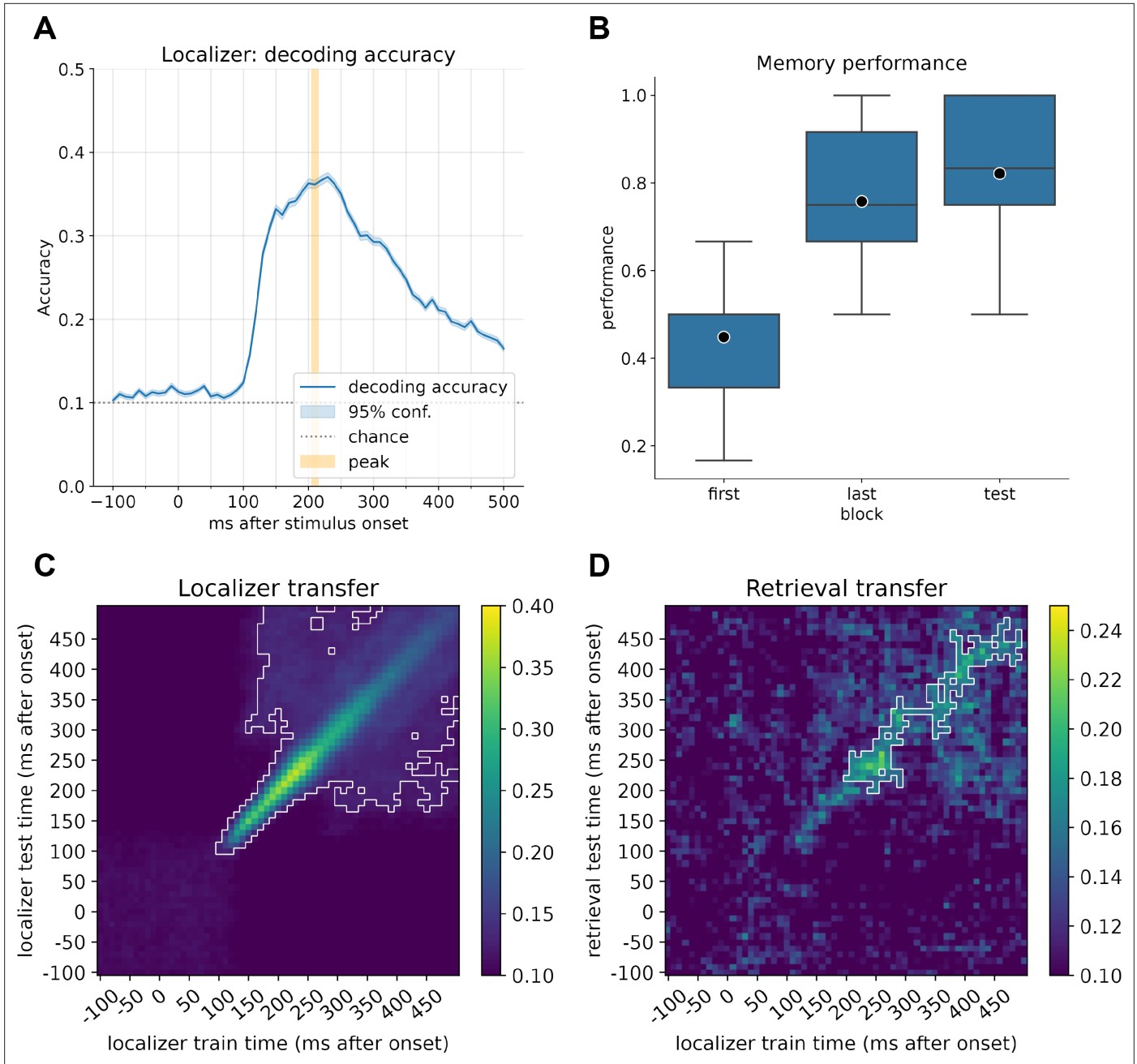

**Figure 1.** Decoder accuracy and learning performance. (**A**) Decoding accuracy of the currently displayed item during the localizer task for participants with a decoding accuracy higher than 30% (n = 21). The mean peak time point across all participants corresponded to 210 ms, with an average decoding peak decoding accuracy of 42% (n = 21). Note that the displayed graph combines accuracies across participants, where peak values were computed on an individual level and then averaged. Therefore, the indicated individual mean peak does not match the average at a group level. (**B**) Memory performance of participants after completing the first block of learning, the last block (blocks 2–6, depending on the speed of learning), and the retrieval performance. (**C**) Classifier transfer within the localizer when trained and tested at different time points determined by cross-validation. (**D**) Classifier transfer from the localizer session to the retrieval session when trained at different time points during training and tested at different time points during cue presentation of the first (predecessor) image cue during retrieval. For (**B**) and (**C**), within the white outline, classification was significantly above chance level (cluster permutation testing, α < 0.05).

The online version of this article includes the following figure supplement(s) for figure 1:

**Figure supplement 1.** Excluded participants based on decoding accuracy and memory performance during retrieval.

**Figure supplement 2.** Number of learning blocks that each participant completed.

*Figure 1 continued on next page*

*Figure 1 continued*

**Figure supplement 3.** During the learning and retrieval blocks, participants were presented two lures next to the correct answer to complete the triplet, one of which was closer to the target and one further away on the graph.

**Figure supplement 4.** Decoding accuracy across time determined by a leave-one-per-class-out cross-validation per participant.

**Figure supplement 5.** Percentage of rejected trials for each participant.

**Figure supplement 6.** Percentage of sensors relevant for each image across all participants (beta weight of sensor location unequal to zero).

maxima across time per permutation. Nevertheless, as we did not have a predefined time lag of interest, and to mitigate multiple comparisons, we additionally computed the mean sequenceness across all computed time lags for each participant (similar to that previously proposed in the context of a sliding-window approach in *Wise et al., 2021*). This measure can help reveal an overall tendency for replay of task states that is invariant to a specific time lag. Our results show that across all participants there is a significant increase in task-related forward sequential reactivation of states (p=0.027, two-sided permutation test with 1000 permutations; 95% of permutation maxima reached at 40–50 ms, *Figure 2B*). Following up on this, in a second analysis, we asked whether mean sequential replay was associated with memory performance and found a significant negative correlation between retrieval performance and forward replay (forward: *r* = −0.46, p=0.031; backward: *r* = −0.13, p=0.56, see *Figure 2C*). In line with previous results (*Wimmer et al., 2020*), low-performing participants had higher forward sequenceness compared to high-performing participants, whose mean sequenceness tended toward zero.

## Closer nodes show stronger reactivation than distant nodes

Next, in a complementary analysis, we asked whether a nonsequential clustered reactivation of items occurs after onset of a cue image (as shown previously for high performers in *Wimmer et al., 2020*). We compared reactivation strength of the two items following the cue image with all items associated

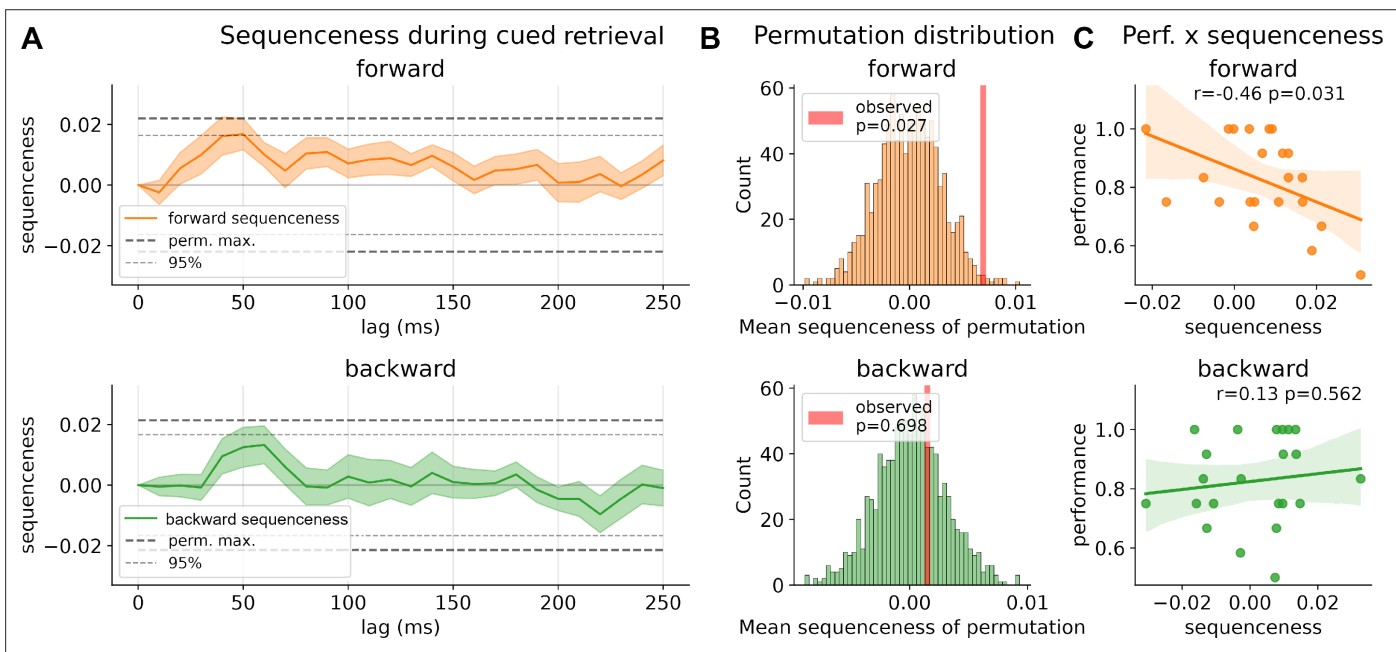

**Figure 2.** Sequenceness during retrieval. (**A**) Strength of forward and backward sequenceness across different time lags up to 250 ms during the 1500 ms window after cue onset. Two significance thresholds are shown: conservative threshold of the maximum of 1000 permutations of classification labels across all time lags and the 95% percentiles (see 'Methods' for details). (**B**) Permutation distribution of mean sequenceness values across 1000 state permutations. Observed mean sequenceness is indicated with a red line. (**C**) Association between memory performance and mean sequenceness value computed across all trials, and time lags, for each participant.

The online version of this article includes the following figure supplement(s) for figure 2:

**Figure supplement 1.** Sequential replay for all learning blocks.

to a distance of more than two steps, subtracting the mean decoded reactivation probabilities from each other. Using this differential reactivation, we found evidence consistent with near items being significantly reactivated compared to items further away within a time window of 220–260 ms after cue onset (*Figure 3A*, p<0.05, permutation test with 10000 shuffles).

To further explore the relation of reactivation strength and graph distances, we analyzed the mean reactivation strength by item distance at peak classifier probabilities and found reactivation strength significantly related to graph distance (repeated-measures ANOVA, $F(4, 80) = 2.98$, p=0.023; *Figure 3B*). When subdividing trials into correct and incorrect responses, we found that this relationship was only significant for trials where a participant successfully retrieved the currently prompted sequence excerpt (repeated-measures ANOVA, $F(4, 80) = 5.0$, p=0.001 for correctly answered trials, *Figure 3C*). For incorrect trials, we found no evidence for this relationship ($F(4, 48) = 1.45$, p=0.230 for incorrectly answered trials), albeit we found no interaction between distance and response type ($F(4, 48) = 1.8$, p=0.13). Note that the last two analyses are based on n = 14 since seven participants had no incorrect trials.

To examine how the 8 min consolidation period affected reactivation, we, post hoc, looked at relevant measures across learning trials in contrast to retrieval trials. For all learning trials, for each participant, we calculated differential reactivation for the same time point we found significant in the previous analysis (220–260 ms). On average, differential reactivation probability increased from pre- to post-resting state; however, the effect was nonsignificant ($t = -1.78$, p=0.08) (*Figure 3D*). Raw mean probabilities between learning and retrieval block for far and distant items are shown in *Figure 3—figure supplement 2*.

## Questionnaire results

Participants were concentrated and alert as indicated by the Stanford Sleepiness Scale (M = 2.3, SD = 0.6, range, 1–3). Participants' summed positive affect score was on average 33.2 (SD = 4.5), and their summed negative affect score was on average 12.2 (SD = 1.9) (PANAS). Individual questionnaire answers for each included participants are available in the supplementary download in the code repository at GitHub.

## Discussion

We combined a graph-based learning task with machine learning to study neuronal events linked to memory retrieval. Participants learned triplets of associated images by trial and error, where these were components of a simple directed graph with 10 nodes and 12 edges. Using machine learning decoding of simultaneously recorded MEG data, we asked what brain processes are linked to retrieval of this learned information and how this relates to the underlying graph structure. We show that learned graph items are retrieved by a simultaneous, clustered, reactivation of items and that the associated reactivation strength relates to graph distances.

Memory retrieval is thought to involve reinstatement of previously evoked item-related neural activity patterns (*Danker and Anderson, 2010*; *Johnson and Rugg, 2007*; *Staresina et al., 2012*). Both spatial and abstract information is purported to be encoded into cognitive maps within the hippocampus and related structures (*Behrens et al., 2018*; *Bellmund et al., 2018*; *Epstein et al., 2017*; *Garvert et al., 2017*; *O'Keefe and Nadel, 1979*; *Peer et al., 2021*). While, for example, spatial distance within cognitive maps is encoded within hippocampal firing patterns (*Theves et al., 2019*), it is unclear how competing, abstract, candidate representations are accessed during retrieval (*Kerrén et al., 2018*; *Kerrén et al., 2022*; *Spiers, 2020*). Two separate mechanisms seem plausible. First, depth-first search might enable inference in not yet fully consolidated cognitive maps by sequential replay of potential candidates (*Mattar and Daw, 2018*; *Nyberg et al., 2022*). Second, breadth-first search could be deployed involving simultaneous activation of candidates when these are sufficiently consolidated within maps that support noninterfering co-reactivation of competing representations (*Mattar and Lengyel, 2022*), or when exhaustive replay would be too expensive computationally. Indeed, consistent with this, *Wimmer et al., 2020* showed that for regular memory performance, sequential and temporally spaced reactivation of items seems to 'piece together' individual elements. This contrasted with high performers who showed a clustered, simultaneous, reactivation profile. We replicate this clustered reactivation and show that its strength reflects distance on a graph structure.

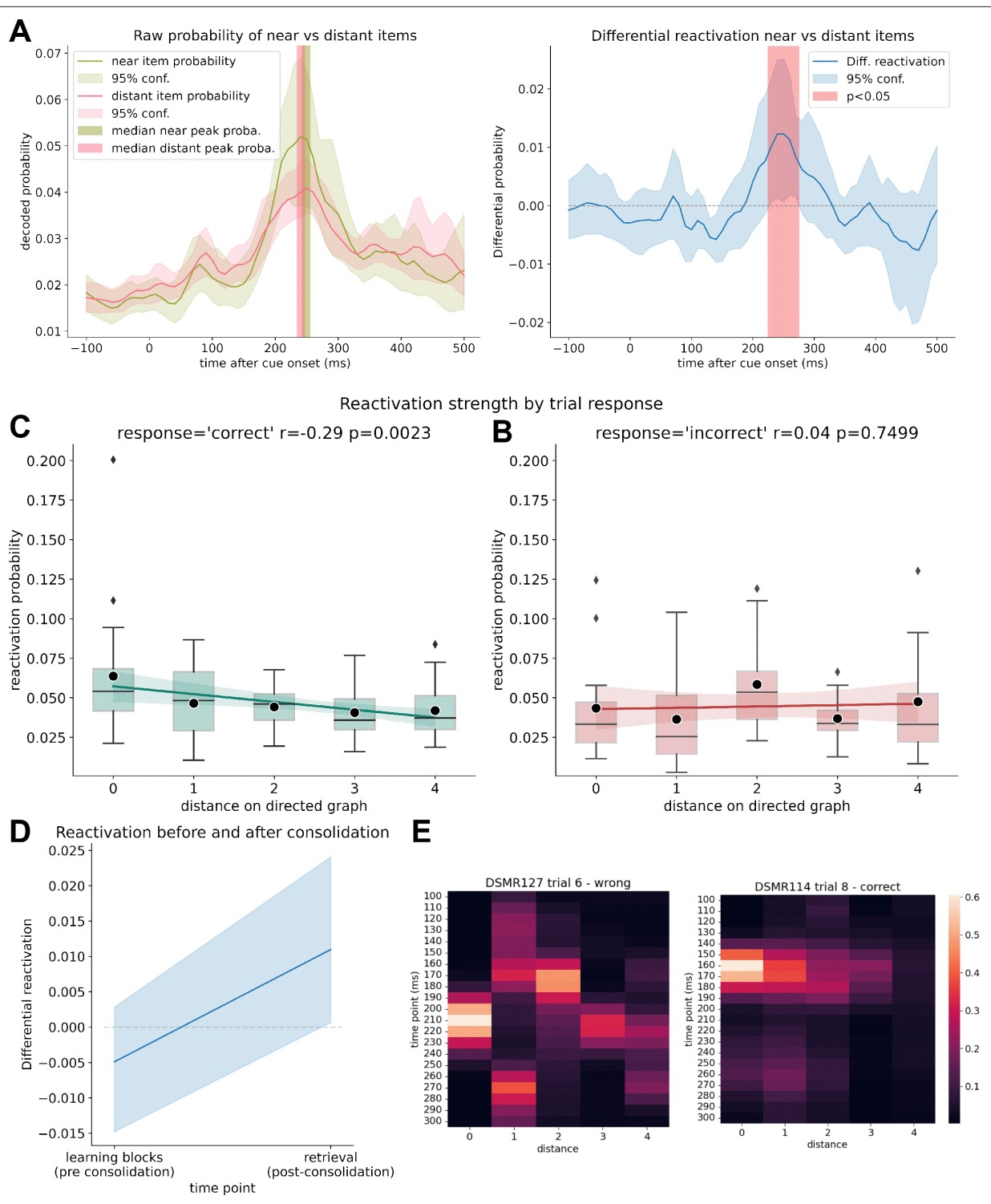

**Figure 3.** Clustered reactivation during retrieval. (**A**) Decoded raw probabilities for off-screen items that were up to two steps ahead of the current stimulus cue ('near') vs. distant items that were more than two steps away on the graph, on trials with correct answers. The median peak decoded probability for near and distant items was at the same time point for both probability categories. Note that the displayed lines reflect the average probability while, to eliminate the influence of outliers, the peak displays the median. (**B**) Differential reactivation probability between off-screen items that were up to two steps ahead of the current stimulus cue vs. distant items that were more than two steps away on the graph for trials with correct answers. Between 220 and 260 ms, the next items are simultaneously reactivated significantly more than the items that are further away (p<0.05; permutation test with 10,000 shuffles). (**C**) Reactivation strength of items after retrieval cue onset by distance of items to the currently on-screen stimulus subdivided into trials in which participants answered correctly (left) and in which participants did not know the correct answer (right). A correlation

*Figure 3 continued on next page*

*Figure 3 continued*

between reactivation strength and distance can only be seen in case of successful retrieval (but see also limitations for a discussion of the low trial and participant number in this sub-analysis). Mean probability values are marked by black dots. (**D**) Mean differential reactivation at peak time point (220–260 ms) during all learning trials (before consolidation) compared to retrieval trials. (**E**) Example activations of a successful retrieval (left) and a failed retrieval (right), sorted by distance to current cue. Colors indicate probability estimates of the decoders.

The online version of this article includes the following figure supplement(s) for figure 3:

**Figure supplement 1.** Mean raw probabilities of near vs far item reactivation at peak time point (210–240 ms, see *Figure 3B*) during learning and retrieval blocks.

**Figure supplement 2.** Reactivation strength of items after retrieval cue onset by distance of items to the currently on-screen stimulus.

This complements previous findings of graded pattern similarity during memory search representing distance within the search space (*Manning et al., 2011*; *Tarder-Stoll et al., 2023*). As this effect was evident only for correct choices, the finding points to its importance for task performance.

According to *Wimmer et al., 2020*, we found that the strength of replay is related to weaker memory performance. This suggests that the expression of sequential replay or simultaneous reactivation depends on the stability of an underlying memory trace. However, we acknowledge that it remains unclear which factors enable recruitment of either of these mechanisms. A crucial step in consolidation encompasses an integration of memory representations into existing networks (*Dudai et al., 2015*; *Sekeres, 2017*). In *Wimmer et al., 2020*, participants had little exposure to the learning material and replay was measured after a substantial retention period that included sleep, where the latter is considered to strengthen and transform memories via repeated replay (*Diekelmann and Born, 2010*; *Feld and Born, 2017*). This contrasts with the current task design, which solely involved several blocks of learning and retrieval and only a relatively brief period of consolidation.

Intriguingly, it has been speculated that retrieval practice may elicit the same transformation of memory traces as offline replay (*Antony et al., 2017*). In line with this reasoning, it is possible that both consolidation during sleep and repeated practice have similar effects on the transformation of memories, and consequently on mechanisms that support their subsequent retrieval. This possibility is especially interesting in the light of retrieval practice enhancing memory performance more than is the case for restudy (*McDermott, 2021*), a finding also in line with evidence that replay during rest prioritizes weakly learned memories (*Schapiro et al., 2018*). It is known that retrieval practice reduces the pattern similarity of competing memory traces in the hippocampus (*Hulbert and Norman, 2015*) and, as in the case of our graph-based task, may enable clustered reactivation since differences in timing of reactivation are no longer required to distinguish correct from incorrect items. Therefore, we speculate that clustered reactivation may be a physiological correlate of retrieval facilitated either by repeated retrieval testing-based learning (as in our study) or sleep-dependent memory consolidation (as in *Wimmer et al., 2020*). This implies that there may be a switch from sequential replay to clustered reactivation corresponding to when learned material can be accessed simultaneously without interference. This suggestion could be systematically investigated by, for example, manipulating retrieval practice, retention interval, and the difficulty of a graph-based task. Nevertheless, even though our results show a nominal, nonsignificant increase in reactivation from learning to retrieval (see *Figure 3D*), due to experimental design features our data do not enable us to test for a hypothesized switch for sequential replay (see also 'Limitations' and *Figure 2—figure supplement 1*). Finally, even though we primarily focused on the mean sequenceness scores across time lags, there appears to be a (nonsignificant) peak at 40–60 ms. While simultaneous forward and backward replay is theoretically possible, we acknowledge that it is somewhat surprising and, given our paradigm, could relate to other factors such as autocorrelations (*Liu et al., 2021a*).

## Limitations

There are limitations to our study, many of which originate from a suboptimal study design that resulted in a relatively limited number of trials for the retrieval session per participant. Additionally, as we performed criteria learning, a sub-group analysis as in *Wimmer et al., 2020* was not feasible as the median performance in our sample was 83% (mean 81%), with six participants exactly at that threshold, resulting in a very high cutoff. Our design also meant participants had different number of learning blocks (2–6 blocks, see *Figure 1—figure supplement 2*), making a comparison of learning progress across participants difficult. While we closely follow the analysis approach taken in *Wimmer*

*et al., 2020*, we did not explicitly preregister the confirmatory analysis of the retrieval data as such. We do acknowledge that only a somewhat limited number of trials were available for analysis, affecting especially the analysis of incorrect answers. In addition, the number of low-performing participants was low in our study, which would render a performance-dependent sub-analysis underpowered. Finally, we want to acknowledge that by selecting a time window for the clustered reactivation we cannot distinguish very fast replay events ($\leq$ 30 ms) from clustered reactivation if they are contained exactly within that specific reactivation analysis time window.

## Conclusion

Our findings support a role for a clustered reactivation mechanism for well-learned items during memory retrieval. When interconnected semantic information is retrieved, the retrieval process seems to resemble a breadth-first search, with items sorted by neural activation strength. Additionally, we find that the presence of sequential replay is related to low memory performance. The likely coexistence of two types of retrieval process, recruited dependent on the participants' learning experience, is an important direction for future research. The use of more complex memory tasks, such as explicitly learned associations of graph networks, should enable a more systematic study of this process. Finally, we suggest that accessing information embedded in a knowledge network may benefit from recruitment of either process, replay or reactivation, on the fly.

## Methods
### Participants

We recruited 30 participants (15 men and 15 women), between 19 and 32 years old (mean age 24.7 y). Inclusion criteria were right-handedness, no claustrophobic tendencies, no current or previously diagnosed mental disorder, nonsmoker, fluency in German or English, age between 18 and 35, and normal or corrected-to-normal vision. Caffeine intake was requested to be restricted for 4 hr before the

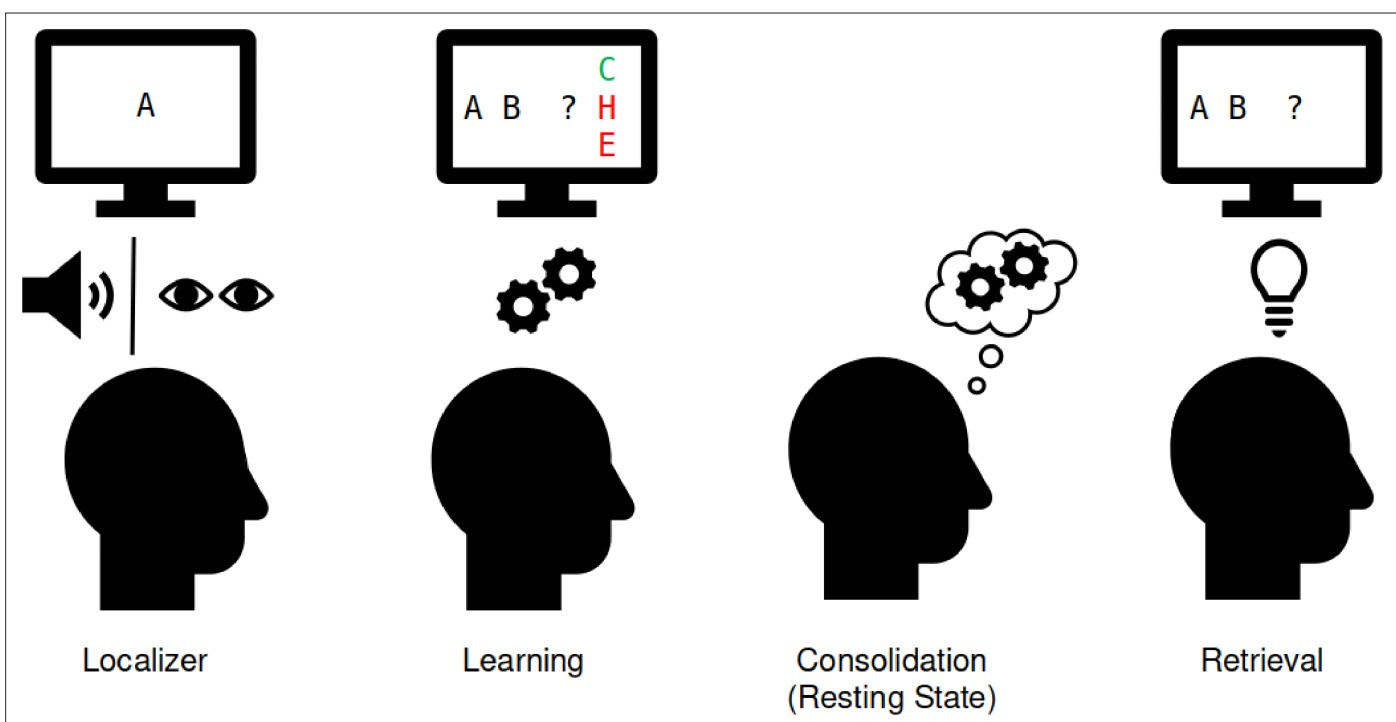

**Figure 4.** Experimental procedure in the magnetoencephalographic (MEG). Localizer task: the 10 individual items were repeatedly presented to the participant auditorily and visually to extract multisensory activity patterns. Learning: participants learned pseudo-randomly generated triplets of the 10 items by trial and error. These triplets were determined by an underlying graph structure. Participants were unaware of the exact structure and graph layout. Consolidation: 8 min of resting state activity were recorded. Retrieval: participants' recall was tested by cueing triplets from a sequence. The letters in the pictograms are placeholders for individual images.

experiment. Participants were recruited through the institute's website and mailing list and various local Facebook groups. A single participant was excluded due to a corrupted data file and replaced with another participant. We acquired written informed consent from all participants, including consent to share anonymized raw and processed data in an open-access online repository. The study was approved by the ethics committee of the Medical Faculty Mannheim of Heidelberg University (ID: 2020-609). While we had preregistered the study design and an analysis approach for the resting state data (https://aspredicted.org/kx9xh.pdf, #68915), here we report analyses of the retrieval period. The current analysis conceptually replicates the analyses and hypotheses of *Wimmer et al., 2020* focusing on the retrieval period albeit in a much more complex and therefore naturalistic paradigm and are therefore, despite not being preregistered, mainly of confirmatory nature. We wish to maintain transparency by acknowledging that the findings from the preregistered analysis concerning the resting state data are being prepared for publication as part of a distinct submission.

## Procedure

Participants came to the laboratory for a single study session of approximately 2.5 hr. After filling out a questionnaire about their general health, their vigilance state (Stanford Sleepiness Scale, *Hoddes et al., 1973*), and mood (PANAS, *Watson et al., 1988*), participants performed five separate tasks while in the MEG scanner. First, an 8 min eyes-closed resting state was recorded. This was followed by a localizer task (~30 min), in which all 10 items were presented 50 times in a pseudo-randomized order, using auditory and visual stimuli. Next, participants learned a sequence of the 10 visual items embedded into a graph structure until they achieved 80% accuracy or reached a maximum of six blocks (7–20 min). Following this, we recorded another 8 min eyes-closed resting state to allow for initial consolidation and, finally, a cued retrieval session (4 min). For an overview see *Figure 4*.

## Stimulus material

Visual stimuli were taken from the colored version (*Rossion and Pourtois, 2001*) of the *Snodgrass and Vanderwart, 1980* stimulus dataset. To increase brain pattern discriminability, images were chosen with a focus on diversity of color, shape, and category (see *Figure 5B*) and for having short descriptive words (one or two syllables) both in German and English. Auditory stimuli were created using the Google text-to-speech API, availing of the default male voice (*SsmlVoiceGender.NEUTRAL*) with the image description labels, either in German or English, based on the participants' language preference. Auditory stimulus length ranged from 0.66 to 0.95 s.

## Task description

### Localizer task

In the localizer task, the 10 graph stimulus items were shown to participants repeatedly in a pseudo-random order, where a DeBruijn sequence (*DeBruijn, 1946*) ensured the number of transitions between any two stimuli was equal. Two runs of the localizer were performed per participant, comprising 250 trials with 25 item repetitions. Each trial started with a fixation cross followed by an inter-trial interval of 0.75–1.25 s. Next, to encourage a multisensory neural representation, the name of the to-be-shown image was played through in-ear headphones (maximum 0.95 s) followed 1.25–1.75 s later by the corresponding stimulus image, shown for 1.0 s. As an attention check, in ~4% of the trials the auditory stimulus did not match the image and participants were instructed to press a button as fast as possible to indicate detection of an incongruent auditory-visual pair. A short break of maximum 30 s was scheduled every 80 trials. Between the two parts of the localizer task, another short break was allowed. Stimulus order was randomized and balanced between subjects. To familiarize the participant with the task, a short exemplar of the localizer task with dummy images was shown beforehand. All subsequent analyses were performed using the visual stimulus onset as a point of reference.

### Graph-learning

The exact same images deployed in the localizer task were randomly assigned to the nodes of the graph, as shown in *Figure 5B*. Participants were instructed to learn a randomized sequence of elements, with the goal of reaching 80% performance within six blocks of learning. During each block, participants were presented with each of the 12 edges of the graph exactly once, in a balanced, pseudo-randomized order. After a fixation cross of 3.0 s, a first image (predecessor) was shown on

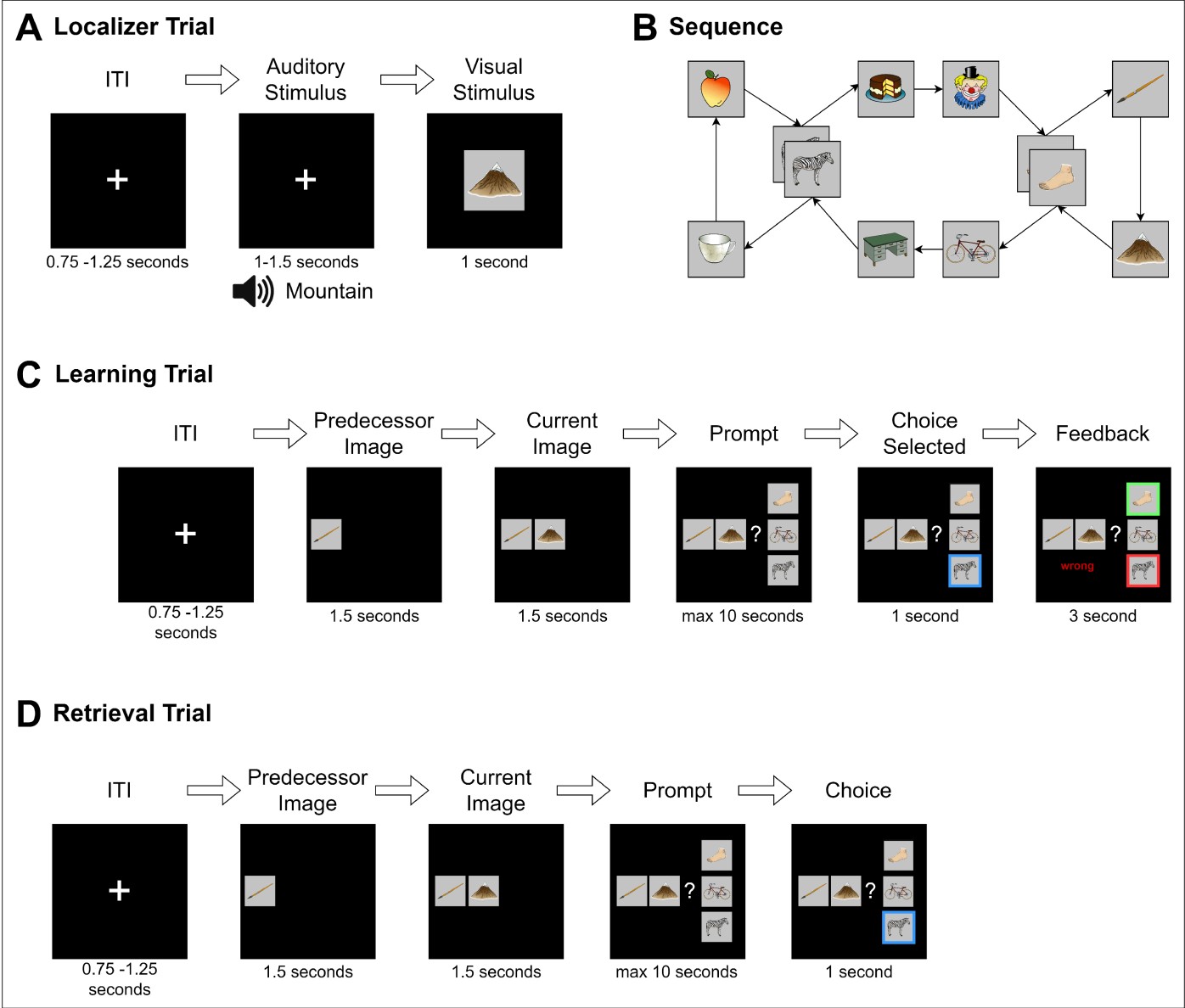

**Figure 5.** Task structure. (**A**) During the localizer task, a word describing the stimulus was played via headphones and the corresponding visual item was then shown to the participant. In 4% of trials, the audio and visual cue did not match, and in this case, participants were instructed to press a button on detection (attention check). (**B**) Graph layout of the task. Two elements could appear in two different triplets. The graph was directed such that each tuple had exactly one successor (e.g., apple→zebra could only be followed by cake and not mug), but individual items could have different successors (zebra alone could be followed by mug or cake). Participants never saw the illustrated birds-eye view. (**C**) During learning, in each trial one node was randomly chosen as the current node. First, its predecessor node was shown, followed by the current node with the participant then given a choice of three items. They were then required to choose the node that followed the displayed cue tuple. Feedback was then provided to the participant. This process was repeated until the participant reached 80% accuracy for any block or reached a maximum of six blocks of learning. (**D**) The retrieval followed the same structure as the learning task, except that no feedback was given.

the left of the screen. After 1.5 s, the second image (current image) appeared in the middle of the screen. After another 1.5 s, three possible choices were displayed in vertical order to the right of the two other images. One of the three choice options was the correct successor of the cued edge. Of the two distractor stimuli, one was chosen from a distal location on the graph (5–8 positions away from the current item), and one was chosen from a close location (2–4 positions away from the current item). Neither of the latter were directly connected to any of the other elements on-screen. Participants used a three-button controller to indicate their answer. The chosen item was then highlighted for 3.0 s, and the participant's performance was indicated ('correct' or 'wrong') (see *Figure 5C*). No

audio was played during learning. The participant was instructed to learn the sequence transitions by trial-and-error, and also instructed that there was no semantic connection between items (i.e., that the sequence did not follow any specific logic related to image content). Participants completed a minimum of two and a maximum of six blocks of learning. To prevent ceiling effects, learning was discontinued if a participant reached 80% accuracy during any block. To familiarize participants with the task, a short example with dummy images was shown before the learning task. Triplets were shown in a random order and choices were displayed in a pseudo-random position that ensured the on-screen position of the correct item could never be at the same position for more than three consecutive trials. Distractor choices were balanced such that exposure to each individual item was approximately equal.

### Resting state

After graph learning, participants completed a resting state session of 8 min. Here, they were instructed to close their eyes and 'to not think of anything particular'. These resting state data are not reported here.

### Retrieval

After the resting state, we presented subjects with a single retrieval session block, which followed the exact layout of the learning task with the exception that no feedback was provided as to whether the entered choices were correct or incorrect (*Figure 5D*).

## MEG acquisition and preprocessing

MEG was recorded in a passively shielded room with a MEGIN TRIUX (MEGIN Oy, Helsinki, Finland) with 306 sensors (204 planar gradiometers and 102 magnetometers) at 1000 Hz with a 0.1–330 Hz band-pass acquisition filter at the ZIPP facility of the Central Institute for Mental Health in Mannheim, Germany. Before each recording, empty room measurements made sure that no ill-functioning sensors were present. Head movement was recorded using five head positioning coils. Bipolar vertical and horizontal electrooculography (EOG) as well as electrocardiography (ECG) was recorded. After recording, the MEGIN proprietary MaxFilter algorithm (version 2.2.14) was run using temporally extended signal space separation and movement correction with the MaxFilter default parameters (*Taulu and Simola, 2006*, raw data buffer length of 10 s, and a subspace correlation limit of 0.98). Bad channels were automatically detected at a detection limit of 7; none had to be excluded. The head movement correction algorithm used 200 ms windows and steps of 10 ms. The HPI coil fit accept limits were set at an error of 5 mm and a g-value of 0.98. Using the head movement correction algorithm, the signals were virtually repositioned to the mean head position during the initial localizer task to ensure compatibility of sensor-level analysis across the recording blocks. The systematic trigger delay of our presentation system was measured and visual stimuli appeared consistently 19 ms after their trigger value was written to the stimulus channel; however, to keep consistency with previous studies that do not report trigger delay, timings in this publication are reported uncorrected (i.e., 'as is', not corrected for this delay).

Data were preprocessed using Python-MNE (version 1.1, *Gramfort et al., 2013*). Data were downsampled to 100 Hz using the MNE function '*resample*' (with default settings, which applies an anti-aliasing filter before resampling with a brick-wall filter at the Nyquist frequency in the frequency domain) and ICA applied using the '*picard*' algorithm (*Ablin et al., 2018*) on a 1 Hz high-pass filtered copy of the signal using 50 components. As recommended, ICA was set to ignore segments that were marked as bad by *Autoreject* (*Jas et al., 2017*) on two-second segments. Components belonging to EOG or ECG and muscle artifacts were identified and removed automatically using MNE functions '*find_bads_eog*', '*find_bads_ecg*', and '*find_bads_emg*', using the EOG and ECG as reference signals. Finally, to reduce noise and drift, data were filtered with a high-pass filter of 0.5 Hz using the MNE filter default settings (hamming window FIR filter, –6 dB cutoff at 0.25 Hz, 53 dB stop-band attenuation, filter length 6.6 s).

Trials from the localizer and retrieval task were created from –0.1 to 0.5 s relative to visual stimulus onset to train decoders. For the sequenceness analysis related to the retrieval, trials were created from 0 to 1.5 s after onset of the second visual cue image. No baseline correction was applied. To detect artifacts, *Autoreject* was applied using default settings, which repaired segments by interpolation in

case artifacts were present in only a limited number of channels and rejected trials otherwise (see *Figure 1—figure supplement 5*). Finally, to improve numerical stability, signals were rescaled to similar ranges by multiplying values from gradiometers by $1e^{10}$ and from magnetometers by $2e^{11}$. These values were chosen empirically by matching histograms for both channel types. As outlier values can have a significant influence on the computations, after rescaling, values that were still above 1 or below –1 were 'cutoff' and transformed to smaller values by multiplying with $1e^{-2}$. Anonymized and maxfiltered raw data are openly available at Zenodo (https://doi.org/10.5281/zenodo.8001755), and code is made public on GitHub (https://github.com/CIMH-Clinical-Psychology/DeSMRRest-clustered-reactivation, copy archived at *CIMH-Clinical-Psychology, 2024*).

## Decoding framework and training

In line with previous investigations (*Kurth-Nelson et al., 2016*; *Liu et al., 2019*; *Wimmer et al., 2020*), we applied LASSO regularized logistic regression on sensor-level data of localizer trials using the Python package Scikit-Learn (*Pedregosa et al., 2011*). Decoders were trained separately for each participant and each stimulus using *liblinear as* a solver with 1000 maximum iterations and an L1 regularization of C = 6. This value was determined based on it giving the best average cross-validated peak accuracy across all participants when searching within the parameter space of C = 1–20 in steps of 0.5 using the same approach as outlined below (note that Scikit-Learn shows stronger regularization with lower C values, opposite to, e.g., MATLAB). To circumvent class imbalance due to trials removed by *Autoreject*, localizer trials were stratified such that they contained an equal number of trials from each stimulus presentation by randomly removing trials from over-represented classes. Using a cross-validation schema (leaving one trial out for each stimulus per fold, i.e., 10 trials left out per fold), for each participant the decoding accuracy was determined across time (*Figure 1A*). During cross-validation, for each fold, decoders were trained on data of each 10 ms time step and tested on leftout data from the same time step. Therefore, decoding accuracy reflects the separability of the stimulus classes by the sensor values for each time step independently. Decoders were trained using a one-vs-all approach, which means that for each class, a separate classifier was trained using positive examples (target class) and negative examples (all other classes) plus null examples (data from before stimulus presentation, see below). This approach allows the decoder to provide independent estimates of detected events for each class.

For each participant, a final set of decoders (i.e., 10 decoders per participant, for each stimulus one decoder) were trained at 210 ms after stimulus onset, a time point reflecting the average peak decoding time point computed for all participants (for individual decoding accuracy plots, see *Figure 1—figure supplement 4*). For the final decoders, data from before the auditory stimulus onset was added as a negative class with a ratio of 1:2, based upon results from previous publications reaching better sensitivity with higher negative class ratio (*Liu et al., 2021a*). Adding null data allows decoders to report low probabilities for all classes simultaneously in the absence of a matching signal and reduces false positives while retaining relative probabilities between true classes. Together with the use of a sparsity constraint on the logistic regression coefficients, this increases the sensitivity of sequence detection by reducing spatial correlations of decoder weights (see also *Liu et al., 2021a*). For a visualization of relevant sensor positions, see *Figure 1—figure supplement 6*.

Decoders were then applied to trials of the retrieval session, starting from the time point of onset of the second sequence cue ('current image') and extending to just prior to onset of the selection prompt (1.5 s). For each trial, this resulted in 10 probability vectors across the trial, one for each item, in steps of 10 ms. These probabilities indicate the similarity of the current sensor-level activity to the activity pattern elicited by exposure to the stimulus and can therefore be used as a proxy for detecting active representations, akin to a representational pattern analysis approach (*Grootswagers et al., 2017*). As a sanity check, we confirmed that we could decode the currently on-screen image by applying the final trained decoders to the first image shown during retrieval (predecessor stimulus, see *Figure 1D*). Note that we only included data from the current image cue, and not from the predecessor image cue, as we assume the retrieval processes differ and should not be concatenated.

## Sequential replay analysis

To test whether individual items were reactivated in sequence at a particular time lag, we applied TDLM (*Liu et al., 2021a*) on the time span after the stimulus onset of the sequence cue ('current

image'). In brief, this method approximates a time-lagged cross-correlation of the reactivation strength in the context of a particular transition pattern, quantifying the strength of a certain activity transition pattern distributed in time. As input for the sequential analysis, we used the raw probabilities of the 10 classifiers corresponding to the stimuli.

Using a linear model, we first estimate evidence for sequential activation of the decoded item representations at different time lags. For each item $i$ at each time lag $\Delta t$ up to 250 ms, we estimated a linear model of form:

$$Y_i = Y(\Delta t) \times \beta_i(\Delta t)$$

where $Y_i$ contains the decoded probability output of the classifier of item $i$ and $Y(\Delta t)$ is simply $Y$ time lagged by $\Delta t$. When solving this equation for $\beta_i(\Delta t)$, we can estimate the predictive strength of $Y(\Delta t)$ for the occurrence of $Y_i$ at each time lag $\Delta t$. Calculated for each stimulus $i$, we then create an empirical transition matrix $T_e(\Delta t)$ that indexes evidence for a transition of any item $j$ to item $i$ at time lag $\Delta t$ (i.e., a 10 × 10 transition matrix per time lag, each column $j$ contains the predictive strength of $j$ for each item $i$ at time lag $\Delta t$). These matrices are then combined with a ground truth transition matrix $T$ (encoding the valid sequence transitions of interest) by taking the Frobenius inner product. This returns a single value $Z_{\Delta t}$ for each time lag, indicating how strongly the detected transitions in the empirical data follow the expected task transitions, which we term 'sequenceness'. Using different transition matrices to depict forward ($T_f$) and backward ($T_b$) replay, we quantified evidence for replay at different time lags for each trial separately. This process is applied to each trial individually, and resulting sequenceness values are averaged to provide a final sequenceness value per participant for each time lag $\Delta t$. To test for statistical significance, we create a baseline distribution by permuting the rows of the transition matrix 1000 times (creating transition matrices with random transitions; identity-based permutation, *Liu et al., 2021a*) and calculate sequenceness across all time lags for each permutation. The null distribution is then constructed by taking the peak sequenceness across all time lags for each permutation.

## Differential reactivation analysis

To test for clustered, nonsequential reactivation, we adopted the approach used in *Wimmer et al., 2020*. Decoders were trained independently for each stimulus, and all decoders reacted to the presentation of any visual stimulus to some extent. By using differences in reactivation between stimuli, this aggregated approach allowed us to examine whether near items are more strongly activated than distant items more closely, thereby quantifying nonsequential reactivation with greater sensitivity. For each trial, the mean probability of the two items following the current on-screen item was contrasted with the mean probability of all items further away by subtraction. We chose to combine the following pairs of items for two reasons: first, this doubled the number of included trials; secondly, using this approach the number of trials for each category ('near' and 'distant') was more balanced. The two items currently displayed on-screen (i.e., predecessor and current image) were excluded. As only a few trials per participant were available for this analysis, the raw probabilities were noisy. Therefore, to address this we applied a Gaussian smoothing kernel (using scipy.ndimage.gaussian_filter with the default parameter of $\sigma = 1$, which corresponds approximately to taking the surrounding time steps in both directions with the following weighting: current time step: 40%, ±1 step: 25%, ±2 step: 5%, ±3 step: 0.5%) to the probability vectors across the time dimension. By shuffling the stimulus labels 1000 times, we constructed an empirical permutation distribution to determine at which time points the differential reactivation of close items was significantly above chance ($\alpha = 0.05$).

## Graph reactivation analysis

To detect whether reactivation strength was modulated by the underlying graph structure, we compared the raw reactivation strength of all items by distance on the directed graph. First, we calculated a time point of interest by computing the peak probability estimate of decoders across all trials, that is, the average probability for each time point of all trials, of all distances except the previous on-screen item. Then, for each participant, for each trial we sorted all nodes based on their distance to the current on-screen item on the directed graph. Again, we smoothed probability values with a Gaussian kernel ($\sigma = 1$) and ignored the predecessor on-screen item. Following this, we evaluated the sorted decoder probabilities at the previously determined peak time point. Using

a repeated-measures ANOVA on the mean probability values per distance per participant, we then estimated whether reactivation strength was modulated by graph distance.

## Exclusions

Replay analysis relies on a successive detection of stimuli where the chance of detection exponentially decreases with each step (e.g., detecting two successive stimuli with a chance of 30% leaves a 9% chance of detecting a replay event). However, one needs to bear in mind that accuracy is a 'winner-takes-all' metric indicating whether the top choice also has the highest probability, disregarding subtle, relative changes in assigned probability. As the methods used in this analysis are performed on probability estimates and not class labels, one can expect that the 30% are a rough lower bound and that the actual sensitivity within the analysis will be higher. Additionally, based on pilot data, we found that attentive participants were able to reach 30% decodability, allowing its use as a data quality check. Therefore, we decided a priori that participants with a peak decoding accuracy of below 30% would be excluded from the analysis (nine participants in all) as obtained from the cross-validation of localizer trials. Additionally, as successful learning was necessary for the paradigm, we ensured all remaining participants had a retrieval performance of at least 50% (see *Figure 1—figure supplement 1*).

## Code availability

The code of the analysis as well as the experiment paradigm and the stimulus material is available at https://github.com/CIMH-Clinical-Psychology/DeSMRRest-clustered-reactivation, copy archived at *CIMH-Clinical-Psychology, 2024*.

## Acknowledgements

This research was supported by an Emmy-Noether research grant awarded to GBF by the DFG (FE1617/2-1) and a project grant by the DGSM as well as a doctoral scholarship of the German Academic Scholarship Foundation, both awarded to SK. Additionally, we want to thank the ZIPP core facility of the Central Institute of Mental Health for their generous support of the study.

## Additional information

### Funding

| Funder | Grant reference number | Author |
|---|---|---|
| Studienstiftung des Deutschen Volkes | PhD Scholarship | Simon Kern |
| Deutsche Forschungsgemeinschaft | FE1617/2-1 | Gordon B Feld |
| German Sleep Research Society | | Simon Kern |

The funders had no role in study design, data collection and interpretation, or the decision to submit the work for publication.

### Author contributions

Simon Kern, Conceptualization, Data curation, Software, Formal analysis, Funding acquisition, Investigation, Visualization, Methodology, Writing - original draft, Project administration, Writing – review and editing; Juliane Nagel, Data curation, Software, Formal analysis, Validation, Writing – review and editing; Martin F Gerchen, Resources, Supervision, Writing – review and editing; Çağatay Gürsoy, Data curation, Formal analysis; Andreas Meyer-Lindenberg, Writing – review and editing; Peter Kirsch, Resources, Writing – review and editing; Raymond J Dolan, Supervision, Methodology, Writing – review and editing; Steffen Gais, Conceptualization, Supervision, Validation, Writing – review and editing; Gordon B Feld, Conceptualization, Supervision, Funding acquisition, Validation, Methodology, Project administration, Writing – review and editing

## Author ORCIDs

Simon Kern ⓘ http://orcid.org/0000-0002-9050-9040
Raymond J Dolan ⓘ http://orcid.org/0000-0001-9356-761X
Gordon B Feld ⓘ https://orcid.org/0000-0002-1238-9493

## Ethics

We acquired written informed consent from all participants, including consent to share anonymized raw and processed data in an open access online repository. The study was approved by the ethics committee of the Medical Faculty Mannheim of Heidelberg University (ID: 2020-609).

Reviewer #1 (Public Review): https://doi.org/10.7554/eLife.93357.4.sa1
Reviewer #2 (Public Review): https://doi.org/10.7554/eLife.93357.4.sa2
Author response https://doi.org/10.7554/eLife.93357.4.sa3

# Additional files

## Supplementary files

• MDAR checklist

## Data availability

MaxFiltered and anonymized MEG raw data as well as behavioural results are available at Zenodo (https://doi.org/10.5281/zenodo.8001755).All code to replicate the analysis is available on GitHub at https://github.com/CIMH-Clinical-Psychology/DeSMRRest-clustered-reactivation, copy achived at *CIMH-Clinical-Psychology, 2024*.

The following dataset was generated:

| Author(s) | Year | Dataset title | Dataset URL | Database and Identifier |
|---|---|---|---|---|
| Gerchen K | 2023 | Reactivation strength during cued recall is modulated by graph distance within cognitive maps | https://doi.org/10.5281/zenodo.8001755 | Zenodo, 10.5281/zenodo.8001755 |

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
