## [Editor Report · eLife assessment]

This magnetoencephalography study reports **important** new findings regarding the nature of memory reactivation during cued recall. It replicates previous work showing that such reactivation can be sequential or clustered, with sequential reactivation being more prevalent in low performers. It adds **convincing** evidence, even though based on limited amounts of data, that high memory performers tend to show simultaneous (i.e., clustered) reactivation, varying in strength with item distance in the learned graph structure. The study will be of interest to scientists studying memory replay.

---

## [Referee Report · Reviewer #1 (Public Review)]

Summary:

Previous work in humans and non-human animals suggests that during offline periods following learning, the brain replays newly acquired information in a sequential manner. The present study uses a MEG-based decoding approach to investigate the nature of replay/reactivation during a cued recall task directly following a learning session, where human participants are trained on a new sequence of 10 visual images embedded in a graph structure. During retrieval, participants are then cued with two items from the learned sequence, and neural evidence is obtained for the simultaneous or sequential reactivation of future sequence items. The authors find evidence for both sequential and clustered (i.e., simultaneous) reactivation. Replicating previous work, low-performing participants tend to show sequential, temporally segregated reactivation of future items, whereas high-performing participants show more clustered reactivation. Adding to previous work, the authors show that an image's reactivation strength varies depending on its proximity to the retrieval cue within the graph structure.

Strengths:

As the authors point out, work on memory reactivation has largely been limited to the retrieval of single associations. Given the sequential nature of our real-life experiences, there is clearly value in extending this work to structured, sequential information. State-of-the-art decoding approaches for MEG are used to characterize the strength and timing of item reactivation. The manuscript is very well written with helpful and informative figures in the main sections. The task includes an extensive localizer with 50 repetitions per image, allowing for stable training of the decoders and the inclusion of several sanity checks demonstrating that on-screen items can be decoded with high accuracy.

Weaknesses:

Of major concern, the experiment is not optimally designed for analysis of the retrieval task phase, where only 4 min of recording time and a single presentation of each cue item are available for the analyses of sequential and non-sequential reactivation. In their revision, the authors include data from the learning blocks in their analysis. These blocks follow the same trial structure as the retrieval task, and apart from adding more data points could also reveal a possible shift from sequential to clustered reactivation as learning of the graph structure progresses. The new analyses are not entirely conclusive, maybe given the variability in the number of learning blocks that participants require to reach the criterion. In principle, they suggest that reactivation strength increases from learning (pre-rest) to final retrieval (post-rest).

On a more conceptual note, the main narrative of the manuscript implies that sequential and clustered reactivation are mutually exclusive, such that a single participant would show either one or the other type. With the analytic methods used here, however, it seems possible to observe both types of reactivation. For example, the observation that mean reactivation strength (across the entire trial, or in a given time window of interest) varies with graph distance does not exclude the possibility that this reactivation is also sequential. In fact, the approach of defining one peak time window of reactivation may bias towards simultaneous, graded reactivation. It would be helpful if the authors could clarify this conceptual point. A strong claim that the two types of reactivation are mutually exclusive would need to be substantiated by further evidence, for instance, a suitable metric contrasting "sequenceness" vs "clusteredness".

On the same point, the non-sequential reactivation analyses use a time window of peak decodability that is determined based on the average reactivation of all future items, irrespective of graph distance. In a sequential forward cascade of reactivations, it could be assumed that the reactivation of near items would peak earlier than the reactivation of far items. In the revised manuscript, the authors now show the "raw" timecourses of item decodability at different graph distances, clearly demonstrating their peak reactivation times, which show convincingly that reactivation for near and far items occurs at very similar time points. The question that remains, therefore, is whether the method of pre-selecting a time window of interest described above could exert a bias towards finding clustered reactivation.

---

## [Referee Report · Reviewer #2 (Public Review)]

Summary:

The authors investigate replay (defined as sequential reactivation) and clustered reactivation during retrieval of an abstract cognitive map. Replay and clustered reactivation were analysed based on MEG recordings combined with a decoding approach. While the authors state to find evidence for both, replay and clustered reactivation during retrieval, replay was exclusively present in low performers. Further, the authors show that reactivation strength declined with an increasing graph distance.

Strengths:

The paper raises interesting research questions, i.e., replay vs. clustered reactivation and how that supports retrieval of cognitive maps. The paper is well written, well structured and easy to follow. The methodological approach is convincing and definitely suited to address the proposed research questions.

The paper is a great combination between replicating previous findings (Wimmer et al. 2020) with a new experimental approach but at the same time presenting novel evidence (reactivation strength declines as a function of graph distance).

What I also want to positively highlight is their general transparency. For example, they pre-registered this study but with a focus on a different part of the data and outlined this explicitly in the paper.

The paper has very interesting findings. However, there are some shortcomings, especially in the experimental design. These are shortly outlined below but are also openly and in detail discussed by the authors.

Weaknesses:

The individual findings are interesting. However, due to some shortcomings in the experimental design they cannot be profoundly related to each other. For example, the authors show that replay is present in low but not in high performers with the assumption that high performers tend to simultaneously reactivate items. But then, the authors do not investigate clustered reactivation (=simultaneous reactivation) as a function of performance due to a low number of retrieval trials and ceiling performance in most participants.

As a consequence of the experimental design, some analyses are underpowered (very low number of trials, n = ~10, and for some analyses, very low number of participants, n = 14).

---

## [Author Response]

The following is the authors’ response to the previous reviews.

**Reviewer #1 (Recommendations For The Authors):**
Results showing reactivation for near and far items separately are now included in Fig. 5 and convincingly suggest a simultaneous reactivation. For me, the open question remaining (see public) review is the degree to which the methods used here to show clustered vs sequential reactivation are mutually exclusive; and if the pre-selection of a time window of peak reactivation (based on all future items) biases the analyses towards clustered reactivation. The discussion would benefit from a brief discussion of these issues.

We have added a brief discussion of the issues. However, we want to clarify a minor point of the public review: While our interpretation implies that replay and reactivation are probably mutually exclusive within a single retrieval event, it does not imply that strategies cannot vary within different retrieval events of the same participant. Nevertheless, we want to address this raised concern (that is, if we understand correctly, that replay events that are contained within the time window of the reactivation analysis could not be distinguished by the chosen methods) and have added it to the discussion.

The corresponding sentence reads:

“[…] Finally, we want to acknowledge that by selecting a time window for the clustered reactivation we cannot distinguish very fast replay events (<=30ms) from clustered reactivation if they are contained exactly within the specific reactivation analysis time window..

**Reviewer #2 (Recommendations For The Authors):**
Figure 5D shows the difference scores between near vs. distant items for learning and retrieval. Similar to Figure 5 from the first version of your paper, the difference score does not show whether reactivation of the near vs. distant items change from learning to retrieval. You could show this change in a 2 (near vs. distant) x 2 (learning vs. retrieval) box plot (corresponding to Figure 5A).

We have added the requested plot as supplement 9 and referred to it in the figure description. However comparing absolute, raw probabilities between different blocks is tricky, as baseline probabilities are varying over time (e.g. due to shift in distance to sensors), therefore, differential reactivation might be better suited as it is a relative measure to compare between blocks.

At the end of the results section, you state: "On average, differential reactivation probability increased from pre to post resting state (Figure 5D).". I would suggest providing some statistical comparison and the corresponding values.

We have calculated and added respective p-value statistics of a T-Test and reported that the increase is only descriptive and not statistically significant.